# Time of urine sampling may influence the association between urine specific gravity and body composition

Patrick B. Wilson[ID][1]*, Brian K. Ferguson[ID][2], Ian P. Winter[3]

1 Associate Professor of Exercise Science, Human Performance Laboratory, School of Exercise Science, Old Dominion University, Norfolk, Virginia, United States of America, 2 Department of Health and Human Performance, Palm Beach Atlantic University, West Palm Beach, Florida, United States of America, 3 Human Performance Laboratory, School of Exercise Science, Old Dominion University, Norfolk, Virginia, United States of America

* pbwilson@odu.edu

## Abstract

Urine specific gravity (USG) is frequently utilized in sports practice and research to assess hydration status. Prior research suggests that individuals with large amounts of fat-free mass (FFM) and muscle have elevated USG, but little is known about whether the time of collection (first-morning vs. spot sampling) and various nutritional factors influence these relationships. This cross-sectional, observational study assessed fasted first-morning (n = 55) and non-fasted spot USG (n = 51) samples in adults and evaluated relationships of USG with body composition and nutrition intake. The InBody 770 was used to estimate FFM, skeletal muscle mass (SMM), and total body water (TBW). Protein, water, and sodium intakes from the 24-hour period before USG assessments were generated based on the Automated Self-Administered 24-hour Recall. Median USG was higher for fasted first-morning samples than non-fasted spot samples (1.018 vs. 1.011, Z = −5.2, p < 0.001). Based on fasted first-morning samples, 41.8% of participants had a USG ≥ 1.020 while the prevalence of USG ≥ 1.020 was 21.6% using non-fasted spot samples. None of the body composition variables (FFM, SMM, TBW) significantly associated with fasted first-morning USG (Spearman ρ < 0.10), while all three variables showed significant, positive associations with non-fasted spot USG (Spearman ρ = 0.32–0.36, p < 0.05). None of the dietary variables were significantly associated with either fasted first-morning or non-fasted spot USG. Although previous research has shown the FFM positively associates with USG, this investigation provides evidence that this relationship could depend on sampling time. Non-fasted spot samples, in comparison to fasted first-morning samples, may be impacted by FFM to a greater degree.

**Data availability statement:** All relevant data are within the manuscript and its Supporting Information files.

**Funding:** The author(s) received no specific funding for this work.

**Competing interests:** The authors have declared that no competing interests exist.

## Introduction

Urine specific gravity (USG) is frequently used in sport and other settings (e.g., research, occupational) to diagnosis hypohydration, often based on a threshold of ≥1.020 from refractometry [1,2]. Despite the common use of this practice, evidence for relying on USG on its own as a means of assessing hydration status remains questioned in the literature [3,4]. Previous literature suggests that USG may be more susceptible to misclassifying hydration status when acute body water fluxes occur, such as when active individuals and athletes experience large fluid losses from sweating [3,5].

Another potential problem with using USG to identify hypohydration is that it may be naturally elevated in individuals with larger body masses [6–9]. In a cross-sectional study of male rugby players and distance runners, Hamouti et al. [6] found significant differences in USG based on body size. The rugby players, who weighed 29 kg more on average than runners, had higher average first-morning USG across six sampling days (1.021 vs. 1.016), and the authors reported that this led to more false positives in assessing hypohydration. A possible reason why larger individuals have elevated USG is that they also have more fat-free mass (FFM) and muscle, body tissues that are major sources of urinary metabolites like creatinine. Baxmann et al. [10] examined a number of urinary metabolites and found stronger correlations between creatinine levels (serum and urine) and lean mass than between creatinine and body weight. In addition, multiple studies have found small-to-moderate positive associations between FFM or skeletal muscle mass (SMM) and USG [6–9]. In totality, current literature tentatively supports the idea that individuals with larger amounts of FFM and SMM may be over-diagnosed with hypohydration when using a set USG threshold of ≥1.020. However, this suggestion requires confirmation given that larger, muscular individuals could be more prone to true hypohydration due to differences in water turnover and sweat losses [11].

Beyond body composition, USG can also be affected by dietary intake. While USG is negatively associated with daily water intake in the literature [5], other dietary factors may influence USG independently of fluid [3]. Dietary protein, for example, seems to affect USG via metabolic byproducts of protein metabolism (e.g., urea) that are excreted in urine [12,13]. As reported by Martin et al. [12], individuals show increased USG when fed high-protein diets. In addition, consumption of electrolytes, particularly sodium, can acutely impact USG [14,15].

Studies have also shown that USG varies based on sampling time of day [3], with first-morning USG values often being higher than spot samples taken later in the afternoon [16]. It is also worth pointing out that first-morning urine sampling is typically recommended for the evaluation of hydration status due to a variety of potentially confounding factors that accompany spot sampling [3]. Whether urine sampling time influences the association between FFM/SMM and USG, however, remains unknown. Among studies that have reported a positive correlation between FFM/SMM and USG, urine sampling times have varied substantially [6,8,9].

To the authors' knowledge, no studies to date have simultaneously examined the contributions of the aforementioned factors (FFM, SMM, protein intake, sodium

intake, fluid intake) to USG levels. In addition, whether these associations differ between fasted first-morning and non-fasted spot samples is an important question to address, as the answer could help provide context for practitioners when they test urine at different times of the day. Thus, the purpose of the present study was to assess both fasted first-morning and later-in-the-day, non-fasted spot USG samples and their relation to body composition and nutritional intakes, particularly of protein, fluid, and sodium. We hypothesized that FFM, SMM, dietary protein intake, and dietary sodium intake would associate positively with USG, while fluid intake would negatively associate with USG.

## Methods

### Participants

Participants were recruited from a large, mid-Atlantic university and the surrounding community. The recruitment period lasted from December 1, 2021 to August 31, 2024. Inclusion criteria included the following: at least 18 years of age, weighing under 500 pounds, self-reported engaging in moderate-to-vigorous physical activity at least 3 days per week, and being free from any urinary tract infection. Additionally, due to contraindications for the specific body composition devices used in this study, participants were excluded based on having an implanted electrical device such as a pacemaker or extreme claustrophobia.

After completing eligibility screening, participants attended an initial meeting (either via Zoom or in-person) where they were informed about the study's procedures, risks, benefits, etc., and signed a consent form approved by the Old Dominion University Institutional Review Board. Sixty individuals consented and enrolled to participate in the study, though five of these individuals did not complete an in-person laboratory visit. An additional four participants completed the morning laboratory visit but did not complete their later-in-the-day spot sample visit. Thus, available sample sizes varied based on laboratory visit; n = 55 for the first-morning visit and n = 51 for the spot sample visit. Participant characteristics, including body composition data, are reported in **Table 1**. The self-reported racial/ethnic make-up of the full sample (n = 55) was 34 white, 7 black or African American, 6 Asian or Pacific Islander, 5 mixed race or other, and 3 Hispanic or Latino.

### Procedures

Prior to the morning laboratory visit, participants were asked to refrain from vigorous or very prolonged (>60 min) exercise and refrain from food/fluid intake for at least 12 and 8 hours, respectively. Participants were also specifically instructed to refrain from urinating prior to arriving to the laboratory. After arrival, participants provided a urine sample that was tested twice using a handheld digital refractometer (PAL-10S, Atago, Japan), and the average of the two values was used. Before each test, the refractometer was tested with distilled water and re-zeroed if distilled water was measured at a value different from 1.000. Then, a sucrose standard of 4 g/ 100 mL water was measured, with values between 1.016 and 1.018 being deemed acceptable [17].

**Table 1. Participant characteristics.**

|  | Fasted first-morning USG (n = 55) | Non-fasted spot USG (n = 51) |
|---|---|---|
| Age (y) | 26 (22-33) | 26 (22-34) |
| Men/ women | 40/ 15 | 37/ 14 |
| Body mass (kg) | 78.9 (67.7-86.7) | 78.0 (66.1-86.7) |
| Fat-free mass (kg) | 64.2 (54.1-69.9) | 63.6 (52.1-70.1) |
| Fat mass (kg) | 14.3 (10.2-20.2) | 14.3 (10.1–20.5) kg |
| Skeletal muscle mass (kg) | 36.7 (30.0-40.4) | 36.0 (29.5-40.5) |
| Total body water (kg) | 47.0 (39.5-51.1) | 46.6 (38.0-51.2) |

Values are reported as median (25th-75th percentile). USG, urine specific gravity.

Upon completion of USG measurement, body composition was measured using air displacement plethysmography (Bod Pod, Life Measurement Instruments, Concord, CA) and bioelectrical impedance (InBody 770, InBody USA, Cerritos, CA). The Bod Pod and InBody 770 both provided estimates of FFM, while the InBody 770 also gave estimates of SMM and total body water (TBW). Estimates of SMM from the InBody 770 and dual-energy x-ray absorptiometry are highly correlated (r > 0.9) with a 2.3-kg mean overestimate of SMM by the InBody 770 [18]. One participant declined to do the Bod Pod test, leaving n = 54 for those data. The FFM estimates from the Bod Pod and InBody 770 were highly correlated (Pearson r = 0.97) and showed a modest absolute difference in FFM (1.3 kg); thus, only estimates from the InBody 770 were used for analyses with USG for the sake of reducing risks of type 1 error.

After body composition testing, participants completed a 24-hour dietary recall utilizing the Automated Self-Administered 24-hour Recall from the National Cancer Institute [19]. This recall tool was used in the present study to document all food and fluid consumed over the past 24 h for the estimation of nutrient intakes, focusing on dietary protein, water, and sodium. This method provided a reasonably accurate estimation of recent protein intake [20]. Participants were also asked to complete a background questionnaire to include demographic information (age, race/ethnicity, sex).

An additional laboratory visit was conducted on a separate day, typically at 12 p.m. or later, for the measurement of spot USG. Two participants completed this visit at a slightly earlier time (between 11 a.m. and noon). The median visit start time was 1:00 p.m., with a range of start times from 11 a.m. to 5:30 p.m. Participants were asked to refrain from vigorous or very prolonged (>60 min) exercise for at least 12 h before this visit; however, there were no restrictions on pre-visit food/fluid intake. Upon arrival, participants reported the approximate time of their last urine void and then provided a urine sample for analysis. Next, participants completed an additional 24-h recall. Due to technical problems, two participants neglected to complete the 24-h dietary recall at this spot sample visit. Body composition testing was not repeated because it is generally consistent across short time intervals in non-dieting adults [21].

## Statistical analysis

The data were analyzed with version 29 of SPSS software (IBM Corp., Armonk, NY, USA). The distribution of variables was analyzed by inspecting histograms and Q-Q plots. Age, body fat (kg), spot USG, and dietary intakes associated with the spot USG visit all showed evidence of a right skew. Median (25th-75th percentile) was used for descriptive statistics to keep the data presentation uniform across all variables. A Wilcoxon-Signed Ranks test was used to compare USG values between fasted first-morning and later-in-the-day, non-fasted spot samples. Spearman's ρ was used to evaluate the associations between variables. In cases where a predictor variable was significantly associated with one USG variable but not the other USG variable, the Fisher r-to-z transformation was used to examine if the correlation coefficients differed significantly from one another. Because Spearman's correlations were used in place of Pearson's correlations, a value of 1.06 was used in the standard error calculation instead of 1.0 [22]. A two-sided alpha <0.05 was considered as the threshold for statistical significance.

## Results

Median (25th-75th percentile) values for USG were 1.018 (1.014–1.023) and 1.011 (1.003–1.018) for the fasted first-morning and non-fasted spot samples, respectively. Among the 51 participants with both samples, fasted first-morning USG was higher than non-fasted spot USG based on a Wilcoxon-Signed Ranks test (Z = −5.2, p < .001). Based on fasted morning samples, 41.8% of participants had a USG ≥ 1.020 while the prevalence of USG ≥ 1.020 was 21.6% using non-fasted spot samples. There was a modest-sized positive correlation between fasted first-morning USG and non-fasted spot USG (n = 51; ρ = 0.39, p = .005). There was also a significant positive correlation (n = 51; ρ = 0.32, p = .021) between later-in-the-day non-fasted spot USG and time duration since last void (i.e., total hours from last pre-visit void to time of spot USG). The median time duration since last void was 2 (1–4) h.

Correlations between body composition data and USG are shown in **Table 2**. None of the body composition variables were associated with fasted first-morning USG, while FFM, SMM, and TBW all showed significant, positive associations with non-fasted spot USG. The relationship between SMM and non-fasted spot USG values is presented visually in **Fig 1**. Based on the Fisher r-to-z transformation, the size of correlations did not significantly differ for SMM (p = .15), FFM (p = .21), and TBW (p = .22) when comparing correlations based on fasted first-morning USG vs. non-fasted spot USG.

Descriptive data for the nutrition intake data are presented in **Table 3**, with correlations between USG and dietary data shown in **Table 4**. None of the variables of interest (protein, water, sodium) were significantly associated with either fasted first-morning or non-fasted spot USG.

Table 2. Correlations between USG and body composition data from the InBody 770.

|  | Fasted first-morning USG (n = 55) | Non-fasted spot USG (n = 51) |
|---|---|---|
| Fat-free mass | 0.08 (.582) | 0.32 (.021)* |
| Fat mass | 0.12 (.386) | −0.13 (.359) |
| Skeletal muscle mass | 0.08 (.551) | 0.36 (.009)** |
| Total body water | 0.07 (.598) | 0.32 (.024)* |

USG, urine specific gravity. * p < .05, **p < .01. Correlation co-efficients are based on Spearman's ρ.
p values are shown in parentheses.

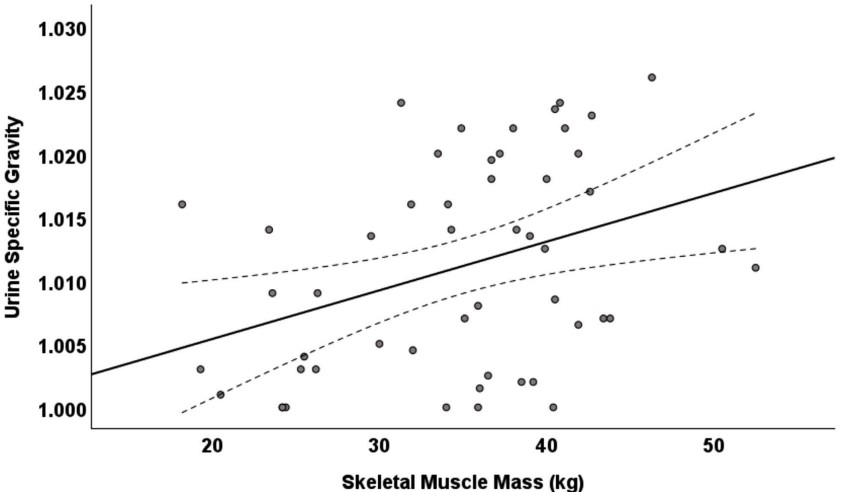

**Fig 1. Scatterplot of the relationship between skeletal muscle mass (SMM) and spot sample USG, with line of best fit and 95% confidence intervals shown.**

Table 3. Nutrient intakes over the 24 h preceding each visit.

|  | Fasted first-morning USG (n = 55) | Non-fasted spot USG (n = 49) |
|---|---|---|
| Energy (kcal) | 2,292 (1,723−2,846) | 2,555 (1,650−3,075) |
| Protein (g) | 115 (76-154) | 148 (85-177) |
| Water (g) | 2,661 (1,600−3,831) | 3,008 (1,519−4,721) |
| Sodium (mg) | 4,134 (2,847−5,652) | 3,979 (3,228−6,405) |

**Table 4. Correlations between USG and dietary intake data.**

|  | Fasted first-morning USG (n = 55) | Non-fasted spot USG (n = 49) |
|---|---|---|
| Protein | −0.21 (.134) | 0.26 (.066) |
| Water | −0.21 (.121) | 0.05 (.711) |
| Sodium | −0.17 (.227) | 0.13 (.391) |

Correlation co-efficients are based on Spearman's ρ. p values are shown in parentheses.

## Discussion

In agreement with several previous investigations, the present study found that measures of body composition—principally FFM and SMM—were positively associated with USG [6,8,9]. This is the first study, at least to the authors' knowledge, to show that these relationships may depend on time of, and the degree of control around, urine sampling (fasted vs. non-fasted). Specifically, the non-fasted spot samples showed a positive association between FFM/SMM and USG, while fasted first-morning samples did not show a significant association between these variables. Other authors have argued that first-morning urine samples, when combined with measurements of body mass and thirst, are more valid measures of hydration status than spot samples due to minimization of confounding factors (diet, activity, etc.) [3]; the findings of the present investigation provide additional rationale for preferentially using fasted first-morning samples over non-fasted spot samples when assessing hydration status. Given the observed association between FFM/SMM and USG from spot samples, misclassification of hydration status may be more likely when individuals of varying sizes and body compositions have their USG tested after the first void of the day. At a minimum, the results of this and other investigations imply that different USG thresholds should be utilized to identify hypohydration in individuals of different body sizes.

Urine sampling times have been variable among prior studies that have found FFM and SMM to associate with USG. Hamouti et al. [6] evaluated USG upon waking across six days in male rugby players and runners and found a moderate-sized correlation between estimated muscle mass and USG ($r = 0.50$, $p = .03$). A later analysis by Wilson [8], which found positive associations between FFM and having a USG ≥ 1.020, was based on spot samples taken at various times throughout the day (morning, afternoon, or evening). Subsequently, Wilson and Winter [9] performed a quantitative review of the literature and reported that FFM had a significant, positive association with USG in both men (n = 91 estimates; $\rho = 0.36$, $p < .001$) and women (n = 22 estimates; $\rho = 0.57$, $p = .006$). However, the studies included in their review utilized a variety of sampling approaches to quantify USG (single value vs. average of values; fasted first-morning vs. non-fasted spot sampling), and their analysis was unable to account for those factors.

It is difficult to say why differences exist between the present study and Hamouti et al. [6] when it comes to the association between SMM and first-morning USG. Sample size and sampling variability may account for at least some of the discrepancy, as Hamouti et al. [6] relied on a significantly smaller sample (n = 18) than the present study. However, even with the larger sample size employed by our study, the difference between correlation coefficients (Fisher r-to-z transformation) for first-morning and spot samples was insignificant, meaning that our findings should be interpreted cautiously. Due to the somewhat inconsistent findings to date, additional research examining the influence of time of day on the connection between SMM and USG is clearly warranted. Time from last void to sample collection should also be examined as a modifying factor given that it showed a positive association ($\rho = 0.32$) with spot USG values in this study.

Our hypotheses related to dietary intakes and USG were not supported by the results. While higher intakes of dietary protein and sodium have been shown to increase USG in some experiments [12,14,15], they did not correlate significantly with either first-morning or spot USG in the present study. Martin et al. [12] revealed that a high-protein diet increased USG relative to a moderate-protein diet, but the amount of protein in the high group (3.6 g/kg/day) was well beyond the protein intake in our study, which may partly explain the discrepant findings. Regarding sodium, ingesting large amounts acutely can impact USG, but the effects depend on the volume of fluid ingested alongside sodium, among other factors

[14]. In addition, less is known about how chronic high-sodium diets impact USG, though a recent experiment found little effect of a 7-day diet supplemented with 3,900 mg of sodium [23].

While not observed in all research [1], measures of fluid intake often correlate with USG in various populations (e.g., [5,24]). However, the relationship between fluid intake and USG is likely to depend on several factors, including the method of assessment for each variable. For example, Perrier et al. [5] found that USG values based on 24-hour urine collections were associated with fluid intake while USG from first-morning samples was not. Other specific reasons for the lack of significant association between fluid intake and USG in our study could include the unknown validity of the water variable from the Automated Self-Administered 24-hour Recall and our modest sample size. Perrier et al. [5] found a non-significant correlation between first-morning USG and total fluid intake that was relatively similar in size ($r = -0.33$) to the observed association between first-morning USG and water intake in this study ($\rho = -0.21$).

There are a few important limitations to this investigation. While the Automated Self-Administered 24-hour Recall is a widely used method to assess recent dietary intake [19], its validity for some nutrients, particularly dietary water, is unknown. In addition, while the estimate of SMM from the InBody 770 has shown promising validity [18,25], it requires further validation in a variety of populations. The sample size of this study was not sufficient for detecting small effects ($\rho < 0.3$), meaning that we cannot rule out the possibility that the dietary variables may have real, albeit minor, associations with USG. Furthermore, we did not directly measure urine metabolites like creatinine, which would have strengthened our argument that the associations between FFM/SMM and USG are likely to be causal in nature.

Future research on this topic could go in several directions. It might be worthwhile, for example, to examine how manipulating various dietary nutrients (protein, carbohydrate, electrolytes, creatine) affects USG, as experimental research on the topic is rather limited. In addition, studies using measurements of muscle mass from more sophisticated and accurate techniques such as magnetic resonance imaging, computed tomography, and dual-energy x-ray absorptiometry may be insightful [26]. Likewise, given that self-reported race and genetic ancestry are related to serum and urine creatine concentrations [27–29], additional research could explore whether these factors moderate the association between dietary nutrients and USG.

## Conclusions

The present study replicates prior literature showing that there are positive associations between FFM/SMM and USG. A novel finding of this work is that the associations between FFM/SMM and USG may vary based on the time of day and degree of control over sampling (fasted vs. non-fasted), which has implications for the use of USG as a measure of hypohydration in clinical and research settings. Practitioners and researchers should bear in mind that, in contrast to fasted first-morning samples, non-fasted spot USG samples may depend more on the body size and composition of the individual being evaluated. Additional research with larger samples should be conducted to replicate these findings and confirm that they extend to other populations and situations.

## Supporting information

**S1 File. Dataset.**
(XLSX)

## Author contributions

**Conceptualization:** Patrick B. Wilson, Brian K. Ferguson.

**Formal analysis:** Patrick B. Wilson, Brian K. Ferguson.

**Investigation:** Patrick B. Wilson, Brian K. Ferguson, Ian P. Winter.

**Methodology:** Patrick B. Wilson, Brian K. Ferguson.

**Supervision:** Patrick B. Wilson.

**Visualization:** Patrick B. Wilson.

**Writing – original draft:** Patrick B. Wilson, Brian K. Ferguson.

**Writing – review & editing:** Patrick B. Wilson, Brian K. Ferguson, Ian P. Winter.

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
