## [Decision Letter · Decision Letter 0]

15 Jul 2025

PONE-D-25-29194Time of urine sampling influences the association between urine specific gravity and body compositionPLOS ONE

Dear Dr. Wilson,

Thank you for submitting your manuscript to PLOS ONE. After careful consideration, we feel that it has merit but does not fully meet PLOS ONE’s publication criteria as it currently stands. Therefore, we invite you to submit a revised version of the manuscript that addresses the points raised during the review process.

 Reviewers found merit in the manuscript but also noted significant improvement was necessary. The methods were a place of particular emphasis (being clearer) as well as the discussion section. Within the discussion, the findings should be interpreted with some of the limitations in mind (as noted by the reviewer). One of the reviewers uploaded a word document with comments, so please make sure you address those too.

We look forward to receiving your revised manuscript.

Kind regards,

Jeremy P Loenneke

Academic Editor

PLOS ONE

2. Please include a separate caption for each figure in your manuscript.

Additional Editor Comments (if provided):

Reviewers' comments:

Reviewer's Responses to Questions

**Comments to the Author**

1. Is the manuscript technically sound, and do the data support the conclusions?

Reviewer #1: Yes

Reviewer #2: Yes

Reviewer #3: Partly

2. Has the statistical analysis been performed appropriately and rigorously? 

Reviewer #1: N/A

Reviewer #2: Yes

Reviewer #3: I Don't Know

3. Have the authors made all data underlying the findings in their manuscript fully available?

Reviewer #1: Yes

Reviewer #2: Yes

Reviewer #3: Yes

4. Is the manuscript presented in an intelligible fashion and written in standard English?

Reviewer #1: Yes

Reviewer #2: Yes

Reviewer #3: Yes

5. Review Comments to the Author

Reviewer #1: Thank you for submitting your research to PLOS ONE. The authors have presented an important topic regarding the time of urine sample and the association between USG and body composition. The authors should be commended for the efforts in conducting the study. Overall, the paper is well written and easy to follow. Please see the attached comments.

Specific comments:

Abstract

Line 22. This is nitpicking but it should be specified what “practice” the authors are referring to. Sports practice?

Introduction

Line 45-46. The threshold number was given, but it might be useful to briefly establish how the urine specific gravity is determined.

Line 54-55. “The rugby players, who weighed 29 kg more than the runner …” This value 29 kg should be an average of 29 kg more than the runner.

Line 56. “… and this led to more false positives in assessing hypohydration.” I like how the authors explain why larger individuals may have elevated USG values and hence false positives, but I was curious whether larger individuals are actually more prone to become a hypohydrated state compared to smaller individuals. If there is no difference between larger and smaller individuals, this information might help lead to lines 62-65.

Line 78-79. “To date, few studies have simultaneously … to USG levels.” Could you cite one or two resources that support this statement?

Line 80-81. “… would be beneficial …” Could the “beneficial” part be more specific and convincing, given that this is the novel aspect of the study (e.g., potential implication)?

Methods

Line 98-99. “… attended an initial visit (either via Zoom or in-person) …” Here, it says that the “initial” visit consisted of paperwork via Zoom or in-person. However, on line 109, it also says “the first laboratory visit,” which makes it sound that the initial visit happened twice. One way to make it clearer is to specifically mention how many visits there were in total.

Line 108-109. “Participants were asked to …” Was there a time range that participants visited the lab for morning measurement?

Line 122. “… leaving n=50 for those data.” I think it should be specifically said that it is for spot sample data.

Line 123-125. The overall justification for using InBody is acceptable, but I am not certain if FFM differences of 1.3 kg are considered small absolute differences.

Line 126-127. “Participants subsequently completed a 24-hour dietary recall …” When exactly is considered “subsequently”? Did it happen the day after? Please specify.

Line 133-134. “An additional visit was conducted on a separate day, typically at 12 p.m. or later.” How many days were given from the previous visit? Was it standardized across individuals? Additionally, what was the time range? Please specify instead of stating 12 pm or later.

Line 139-140. “Participants were not required to be in a fasted state or to restrict fluid intake beforehand.” I think this should be highlighted as a limitation when another condition has restrictions on food and fluid intake (especially the discussion also surrounds the argument that USG is sensitive to dietary intake and body water flux-inducing activities). What about exercise restrictions? Did you also not control that? If so, I think that should be stated in here and the limitation.

Statistical analysis and Results

I would check with a statistician, but if one of the questions of interest is to know whether the relationship between body composition data and USG differs between morning and afternoon, a suitable analysis would be to run a mixed effects model instead of running two separate correlations (which does not directly test whether these correlations differ between times) so you can test whether the relationship is moderated by a different measurement time.

Since the time duration since the last void was used in the analysis, presenting the descriptive values would be informative.

Discussion

Line 185-187. “; the findings of the present investigation provide additional rationale for preferentially using first-morning samples over spot samples when assessing hydration status.” This idea of the first morning being better was not highlighted in the introduction and suddenly appears here. Going back to my previous comment in the introduction about highlighting why it’s “beneficial” to study different time points, maybe the authors might bring up this point in the introduction so the readers have some ideas that it has been suggested that morning measurements might be better. And then you can tie here that this study directly compared within the same study, which is different from references #6, 8, and 9.

This might be too much to add but that reference 3 also stated that “The best practical means of monitoring day-to-day avoidance of dehydration should combine first morning urine concentration (i.e., color) with body mass (weight) and thirst …” Would this be something to add to highlight what’s been recommended?

Line 207-208. “Time from last void to sample collection should also be examined as a modifying factor …” This could also be tested in the current data set if the time duration of the last void is put as a moderator and see if the relationship is moderated by this factor.

Line 209-218. The focus was protein and sodium intake, but have other macronutrients been looked into for association before? I’m coming from the effects of water retention on carbohydrates (which might not be relevant at all).

What would be the future direction of this dietary intake? Is the acute manipulation of those nutrient intakes still worth investigating? Having brief sentences to discuss the future direction might be helpful.

Reviewer #2: I see the benefit for filling a gap in the literature for the focus of this research project. There is a need to define or delineate the effects of USG timing in relation to FFM/SMM. The strengths of the study include addressing a methodology gap, utilizing objective measures for assessing body comp/nutrition data, identifying how hydration status may be inaccurate due to anthropometric factors, and clarity on statistics. The weaknesses are sample size, lack of metabolite measurement data, nutrition assessment tool, lack of clarity regarding health status and/or age/race/gender/etc., and clarity on timing of spot samples.

Reviewer #3: See attached document that utilizes "Track Changes" and are included in the word document. I am hopeful that this will makes it much more efficient for the revisionary process currently and as you proceed. Thanks

6. PLOS authors have the option to publish the peer review history of their article (what does this mean? ). If published, this will include your full peer review and any attached files.

**Do you want your identity to be public for this peer review?** For information about this choice, including consent withdrawal, please see our Privacy Policy .

Reviewer #1: No

Reviewer #2: No

Reviewer #3: No

---

## [Author Response · Author response to Decision Letter 1]

4 Aug 2025

Reviewer #1

Thank you for submitting your research to PLOS ONE. The authors have presented an important topic regarding the time of urine sample and the association between USG and body composition. The authors should be commended for the efforts in conducting the study. Overall, the paper is well written and easy to follow. Please see the attached comments.

• Thank you for the positive comments. We have done our best to incorporate your feedback into the revised version of the manuscript.

Specific comments:

Abstract

Line 22. This is nitpicking but it should be specified what “practice” the authors are referring to. Sports practice?

• Point taken. We have changed the text to ‘sports practice’.

Introduction

Line 45-46. The threshold number was given, but it might be useful to briefly establish how the urine specific gravity is determined.

• We have added text that indicates refractometry is typically used.

Line 54-55. “The rugby players, who weighed 29 kg more than the runner …” This value 29 kg should be an average of 29 kg more than the runner.

• We have edited the text as suggested.

Line 56. “… and this led to more false positives in assessing hypohydration.” I like how the authors explain why larger individuals may have elevated USG values and hence false positives, but I was curious whether larger individuals are actually more prone to become a hypohydrated state compared to smaller individuals. If there is no difference between larger and smaller individuals, this information might help lead to lines 62-65.

• That is a valid and insightful question from the reviewer. There is some debate over whether the elevated USG in larger individuals is due to false positives or actually being more prone to hypohydration. We have discussed this debate in a related paper that looked at the relationships between race/ethnicity, LBM/FFM, and USG (link below). In that paper, we wrote, “In adults, fat-free mass is typically comprised of 70–75% water [31], meaning that those with greater amounts of LBM could theoretically require greater water intakes to maintain their body water stores….However, other research has shown that the variance in daily water turnover is not explained well by anthropometric variables like weight, height, or body mass index [32], and the necessity of taking body size into consideration when making fluid intake recommendations for adults is uncertain [31].” In the present paper, we have amended the text as follows: “However, this suggestion requires confirmation given that larger, muscular individuals could be more prone to true hypohydration due to differences in water turnover and sweat losses [11].”

• https://journals.plos.org/plosone/article?id=10.1371/journal.pone.0304803

Line 78-79. “To date, few studies have simultaneously … to USG levels.” Could you cite one or two resources that support this statement?

• We have edited the text to indicate that we are not aware of any published studies that have simultaneously looked at this combination of factors (FFM, protein intake, sodium intake, fluid intake).

Line 80-81. “… would be beneficial …” Could the “beneficial” part be more specific and convincing, given that this is the novel aspect of the study (e.g., potential implication)?

• Thanks for the suggestion. We have edited the text as follows: “In addition, whether these associations differ between fasted first-morning and non-fasted spot samples is an important question to address, as the answer could help provide context for practitioners when they test urine at different times of the day.”

Methods

Line 98-99. “… attended an initial visit (either via Zoom or in-person) …” Here, it says that the “initial” visit consisted of paperwork via Zoom or in-person. However, on line 109, it also says “the first laboratory visit,” which makes it sound that the initial visit happened twice. One way to make it clearer is to specifically mention how many visits there were in total.

• We have edited the text in both sections to hopefully clarify the various meetings and visits that were involved. There were always three meetings / visits. The first was a consent meeting, the second was the fasted morning lab visit, and the third was the unfasted spot sample visit. Participants could elect to complete the consent visit either online or in person.

Line 108-109. “Participants were asked to …” Was there a time range that participants visited the lab for morning measurement?

• All visits occurred before 12 pm, with the vast majority occurring before 10 am.

Line 122. “… leaving n=50 for those data.” I think it should be specifically said that it is for spot sample data.

• This was an error on our part. There is n=54 for the Bod Pod data (n=55 minus one participant who declined the Bod Pod test). We have edited the text accordingly.

Line 123-125. The overall justification for using InBody is acceptable, but I am not certain if FFM differences of 1.3 kg are considered small absolute differences.

• That’s a fair point. We have changed the language from ‘small’ to ‘modest’. We are open to other wording suggestions from the reviewer.

Line 126-127. “Participants subsequently completed a 24-hour dietary recall …” When exactly is considered “subsequently”? Did it happen the day after? Please specify.

• The recall was carried out in the laboratory after the body composition testing. This has been clarified in the text.

Line 133-134. “An additional visit was conducted on a separate day, typically at 12 p.m. or later.” How many days were given from the previous visit? Was it standardized across individuals? Additionally, what was the time range? Please specify instead of stating 12 pm or later.

• Thanks for the suggestion. We have added the following text regarding the visit start times. “The median visit start time was 1:00 p.m. with range of start times from 11 a.m. to 5:30 p.m.” We did not standardize the number of days between the visits. The number of days was based on individual scheduling with participants.

Line 139-140. “Participants were not required to be in a fasted state or to restrict fluid intake beforehand.” I think this should be highlighted as a limitation when another condition has restrictions on food and fluid intake (especially the discussion also surrounds the argument that USG is sensitive to dietary intake and body water flux-inducing activities). What about exercise restrictions? Did you also not control that? If so, I think that should be stated in here and the limitation.

• This is an interesting point from the reviewer. We don’t necessarily think that the lack of standardization around food and fluid intake for this visit is a limitation. In most studies and in the field/practice, spot USG samples are typically taken and analyzed with no restrictions around fluid intake. Thus, our procedures for this visit mimic what is typically done in practice and thus better generalize to how USG is utilized in the real world. Regarding instructions around exercise, the same instructions were provided to participants for this visit as the other visit. We have added the following text: “Participants were asked to refrain from vigorous or very prolonged (>60 min) exercise for at least 12 h before this visit; however, there were no restrictions on pre-visit food/fluid intake.”

Statistical analysis and Results

I would check with a statistician, but if one of the questions of interest is to know whether the relationship between body composition data and USG differs between morning and afternoon, a suitable analysis would be to run a mixed effects model instead of running two separate correlations (which does not directly test whether these correlations differ between times) so you can test whether the relationship is moderated by a different measurement time.

• Thanks for the suggestion. We have added some additional analyses to compare the correlation coefficients (the Fisher r-to-z transformation). Although the p value for SMM was relatively small (p=.15), it was not statistically significant. These results have been added to the manuscript. We have also added some text to the discussion section in which we re-iterate that further research on the topic is needed and that our results should be interpreted cautiously.

Since the time duration since the last void was used in the analysis, presenting the descriptive values would be informative.

• We appreciate the suggestion. We have added the following text to the results section. “The median time duration since last void was 2 (1-4) h.”

Discussion

Line 185-187. “; the findings of the present investigation provide additional rationale for preferentially using first-morning samples over spot samples when assessing hydration status.” This idea of the first morning being better was not highlighted in the introduction and suddenly appears here. Going back to my previous comment in the introduction about highlighting why it’s “beneficial” to study different time points, maybe the authors might bring up this point in the introduction so the readers have some ideas that it has been suggested that morning measurements might be better. And then you can tie here that this study directly compared within the same study, which is different from references #6, 8, and 9.

• Thanks for the suggestion. We have expanded the second-to-last paragraph of the introduction section as follows: “Studies have also shown that USG varies based on sampling time of day [3], with first-morning USG values often being higher than spot samples taken later in the afternoon [16]. It is also worth pointing out that first-morning urine sampling is typically recommended for the evaluation of hydration status due to a variety of potentially confounding factors that accompany spot sampling [3]. Whether urine sampling time influences the association between FFM and USG, however, remains unknown. Among studies that have reported a positive correlation between FFM/muscle mass and USG, urine sampling times have varied substantially [6,8,9].” We have also edited one sentence in the final paragraph of the introduction to the following: “In addition, whether these associations differ between fasted first-morning and non-fasted spot samples is an important question to address, as the answer could help provide context for practitioners when they test urine at different times of the day.”

This might be too much to add but that reference 3 also stated that “The best practical means of monitoring day-to-day avoidance of dehydration should combine first morning urine concentration (i.e., color) with body mass (weight) and thirst …” Would this be something to add to highlight what’s been recommended?

• Good point. We have edited the text as follows: “Other authors have argued that first-morning urine samples, when combined with measurements of body mass and thirst, are more valid measures of hydration status than spot samples due to minimization of confounding factors (diet, activity, etc.) [3].”

Line 207-208. “Time from last void to sample collection should also be examined as a modifying factor …” This could also be tested in the current data set if the time duration of the last void is put as a moderator and see if the relationship is moderated by this factor.

• See previous comment. We elected to use the Fisher r-to-z transformation to compare the size of correlation co-efficients.

Line 209-218. The focus was protein and sodium intake, but have other macronutrients been looked into for association before? I’m coming from the effects of water retention on carbohydrates (which might not be relevant at all).

• Yes, increasing carbohydrate can certainly lead to fluid retention due to increases intracellular water storage (e.g., https://journals.physiology.org/doi/full/10.1152/japplphysiol.00126.2016). However, we focused primarily on dietary nutrients that are excreted in the urine and can therefore impact USG. Dietary carbohydrate, in the absence of a metabolic disease like diabetes, is not excreted in the urine in meaningful amounts.

What would be the future direction of this dietary intake? Is the acute manipulation of those nutrient intakes still worth investigating? Having brief sentences to discuss the future direction might be helpful.

• Thank you for the suggestion. We have added the following paragraph to the end of the discussion section. “Future research on this topic could go in several directions. It might be worthwhile, for example, to examine how manipulating various dietary nutrients (protein, carbohydrate, electrolytes, creatine) affects USG, as experimental research on the topic is rather limited. In addition, studies using measurements of muscle mass from more sophisticated and accurate techniques such as magnetic resonance imaging, computed tomography, and dual-energy x-ray absorptiometry may be insightful [26]. Likewise, given that self-reported race and genetic ancestry are related to serum and urine creatine concentrations [27-29], additional research could explore whether these factors moderate the association between dietary nutrients and USG.”

Reviewer #2

I see the benefit for filling a gap in the literature for the focus of this research project. There is a need to define or delineate the effects of USG timing in relation to FFM/SMM. The strengths of the study include addressing a methodology gap, utilizing objective measures for assessing body comp/nutrition data, identifying how hydration status may be inaccurate due to anthropometric factors, and clarity on statistics. The weaknesses are sample size, lack of metabolite measurement data, nutrition assessment tool, lack of clarity regarding health status and/or age/race/gender/etc., and clarity on timing of spot samples.

• We appreciate the reviewer’s comments and the time and energy they spent reviewing our manuscript.

Reviewer #3

See attached document that utilizes "Track Changes" and are included in the word document. I am hopeful that this will makes it much more efficient for the revisionary process currently and as you proceed. Thanks

• We have responded to each of the reviewer’s comments within the document.

---

## [Decision Letter · Decision Letter 1]

16 Sep 2025

Time of urine sampling may influence the association between urine specific gravity and body composition

PONE-D-25-29194R1

Dear Dr. Wilson,

We’re pleased to inform you that your manuscript has been judged scientifically suitable for publication and will be formally accepted for publication once it meets all outstanding technical requirements.

Kind regards,

Jeremy P Loenneke

Academic Editor

PLOS ONE

Additional Editor Comments (optional):

I appreciate the comments from all of the reviewers. One remained concerned about the reporting of the different times of the sampling; however, I feel the authors have addressed this in their revised manuscript and response to reviewers. While more could be done in future studies to control for additional variables such as gender, I feel it is outside the scope of this study

Reviewer #1:

Reviewer #2:

Reviewer #3:

Reviewers' comments:

Reviewer's Responses to Questions

**Comments to the Author**

1. If the authors have adequately addressed your comments raised in a previous round of review and you feel that this manuscript is now acceptable for publication, you may indicate that here to bypass the “Comments to the Author” section, enter your conflict of interest statement in the “Confidential to Editor” section, and submit your "Accept" recommendation.

Reviewer #1: All comments have been addressed

Reviewer #2: All comments have been addressed

Reviewer #3: (No Response)

2. Is the manuscript technically sound, and do the data support the conclusions?

Reviewer #1: Yes

Reviewer #2: Yes

Reviewer #3: Partly

3. Has the statistical analysis been performed appropriately and rigorously? 

Reviewer #1: Yes

Reviewer #2: Yes

Reviewer #3: No

4. Have the authors made all data underlying the findings in their manuscript fully available?

Reviewer #1: Yes

Reviewer #2: Yes

Reviewer #3: Yes

5. Is the manuscript presented in an intelligible fashion and written in standard English?

Reviewer #1: Yes

Reviewer #2: Yes

Reviewer #3: Yes

6. Review Comments to the Author

Reviewer #1: I have no further comments for the manuscript. Thank you for thoroughly addressing all of my questions and comments.

Reviewer #2: I see how the article is attempting to fill a ‘gap in the literature’ regarding the timing of USG collection in relation to FFM/SMM. The majority of the items I listed as significantly weakening the impact of the article have been clarified. At least to the level the authors can, which is based upon controlled and uncontrolled factors. I feel this article is impactful, in it's present shape, to help guide future researchers towards an area within the field that could help further this line of research.

Reviewer #3: I appreciate your efforts to improve these works. While it reads more clearly and the authors where able to support their findings in the discussion to some degree, the limitations within the methods to actually address the "time" of day and control relevant variables still need to be addressed.

7. PLOS authors have the option to publish the peer review history of their article (what does this mean? ). If published, this will include your full peer review and any attached files.

**Do you want your identity to be public for this peer review?** For information about this choice, including consent withdrawal, please see our Privacy Policy .

Reviewer #1: No

Reviewer #2: No

Reviewer #3: No

---

## [Editor Report · Acceptance letter]

PONE-D-25-29194R1

PLOS ONE

Dear Dr. Wilson,

I'm pleased to inform you that your manuscript has been deemed suitable for publication in PLOS ONE. Congratulations! Your manuscript is now being handed over to our production team.

Kind regards,

on behalf of

Dr. Jeremy P Loenneke

Academic Editor

PLOS ONE